# High-Fidelity and Long-Duration Human Image Animation with Diffusion Transformer

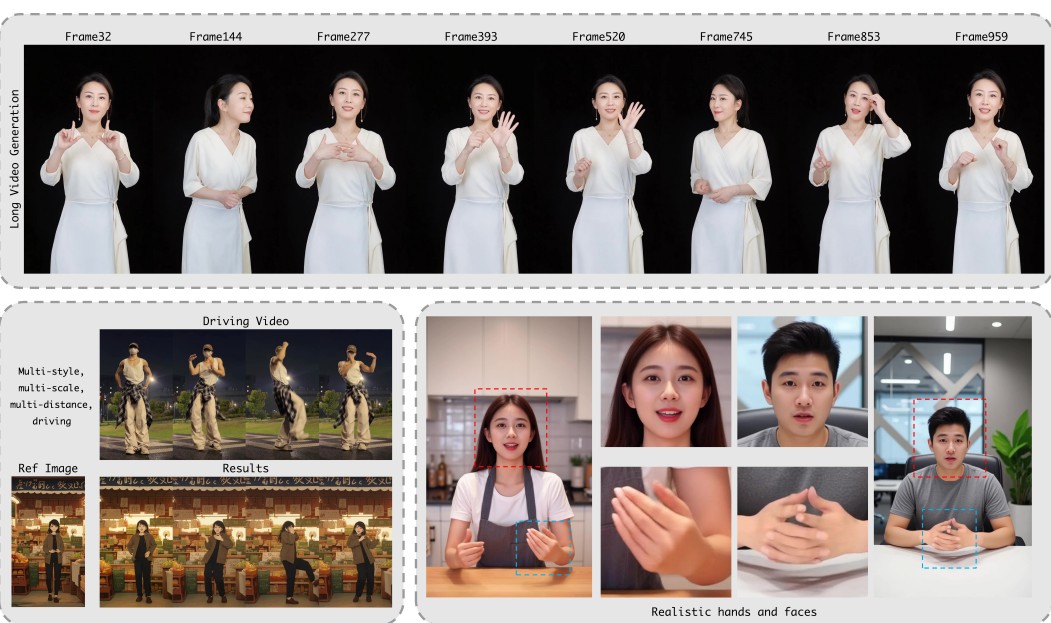

Figure 1: We present a DiT-based framework for generating high-fidelity and long-duration human videos that effectively addresses key challenges related to temporal coherence in long sequences and realistic, plausible hand and face details. To the best of our knowledge, this is the first method to demonstrate animated video examples exceeding one minute in length.

## Abstract

Recent progress in diffusion models has significantly advanced the field of human image animation. While existing methods can generate temporally consistent results for short or regular motions, significant challenges remain, particularly in generating long-duration videos. Furthermore, the synthesis of fine-grained facial and hand details remains under-explored, limiting the applicability of current approaches in real-world, high-quality applications. To address these limitations, we propose a diffusion transformer (DiT)-based framework which focuses on generating high-fidelity and long-duration human animation videos. First, we design a set of hybrid implicit guidance signals and a sharpness guidance factor, enabling our framework to additionally incorporate detailed facial and hand features as guidance. Next, we incorporate the time-aware position shift fusion module, modify the input format within the DiT backbone, and refer to this mechanism as the Position Shift Adaptive Module, which enables video generation of arbitrary length. Finally, we introduce a novel data augmentation strategy and a skeleton alignment model to reduce the impact of human shape variations across different identities. Experimental results demonstrate that our method outperforms existing state-of-the-art approaches, achieving superior performance in both high-fidelity and long-duration human image animation.

# 1 INTRODUCTION

In recent years, the field of visual generation has witnessed remarkable advancements, largely due to the rapid evolution of diffusion models Rombach et al. (2022); Esser et al. (2024); Blattmann et al. (2023); Liu et al. (2024); Kong et al. (2024); Wan et al. (2025). Consequently, breakthroughs in generative diffusion models have propelled the field of human image animation Xue et al. (2024a) to new heights. As a rapidly evolving research area, human image animation has garnered significant interest from both academia and industry. These advancements have opened up a wide range of practical applications, including online retail, digital content creation, and virtual communication platforms.

Human image animation aims to generate lifelike videos from a source image guided by pose sequences. Diffusion-based methods fall into two categories based on backbone architecture: U-Net-based Rombach et al. (2022) and DiT-based Peebles & Xie (2023). U-Net approaches Hu (2024); Zhou et al. (2024); Wang et al. (2024b); Zhang et al. (2024); Li et al. (2024) typically condition on skeletons or depth maps using pretrained Stable Diffusion, but often produce blurry or inconsistent facial and hand details due to architectural limitations. Recently, DiT-based models Luo et al. (2025); Wang et al. (2025b); Zhou et al. (2025), which excel in image and video generation, have been adopted for improved fidelity. However, existing methods still struggle with temporal coherence in long-duration synthesis—especially at clip boundaries—and often fail to generate high-fidelity hand and facial details required for real-world, high-quality applications.

In this paper, we focus on high-fidelity, long-duration, and temporally consistent human animation. Specifically, we adopt the mainstream DiT architecture Wan2.1 Wan et al. (2025) as our foundational framework, and design a hybrid set of implicit guidance signals along with a sharpness guidance factor. These signals include facial and hand-related latent codes for generating expressive faces and realistic hands, respectively. The sharpness guidance factor enables clear hand texture synthesis even under motion blur in the driving video. To ensure long-term temporal consistency, we introduce the Position Shift Adaptive Module, which incorporates the time-aware position shift fusion module proposed by Sonic Ji et al. (2025) and further modifies the input structure for video frames within the DiT backbone. Finally, to mitigate the impact of human shape variations across different identities, we propose a novel data augmentation strategy that improves the robustness of the backbone model to such variations. We also introduce a skeleton alignment model designed to adjust the skeletal structure of the driving subject to match that of the reference identity. Our key contributions are:

- We propose a DiT-based framework and design a hybrid set of implicit guidance signals along with a sharpness guidance factor for high-fidelity human animation, excelling in hand details and motion fidelity.

- We modify the input structure for video frames and employ a latent code shift strategy to enable unlimited duration generation while maintaining temporal coherence.

- We propose a data augmentation strategy and introduce an alignment model to mitigate the impact of human shape variations across different identities.

# 2 RELATED WORK

## 2.1 U-NET-BASED HUMAN IMAGE ANIMATION

Human image animation has gained significant attention since the rise of GANs Goodfellow et al. (2020), with extensive prior work Wang et al. (2018); Liu et al. (2019); Siarohin et al. (2019). Recently, latent diffusion models have pushed the field forward: DisCo Wang et al. (2024a) uses CLIP Radford et al. (2021) for appearance encoding and ControlNet Zhang et al. (2023) for background preservation; Animate Anyone Hu (2024) introduces a ReferenceNet and temporal attention for improved consistency. More recent approaches leverage depth and 3D representations (e.g., SMPL Loper et al. (2023), HaMeR Pavlakos et al. (2024)) for better control Zhu et al. (2024); Zhou et al. (2024); Xue et al. (2024b). Despite these advances, generating photorealistic animations remains challenging, as structural distortions and pose misalignments often persist even with accurate driving inputs.

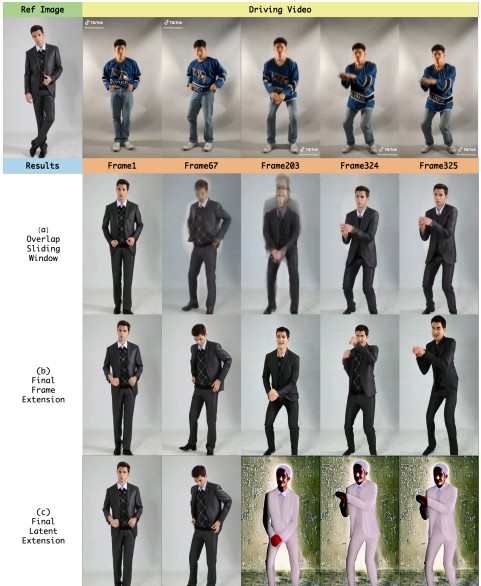

Figure 2: UniAnimate-DiT results using different long video generation strategies: (a) overlapped sliding window; (b) final frame extension; (c) final latent extension.

## 2.2 DiT-BASED HUMAN IMAGE ANIMATION

Recent work has adapted video diffusion transformers to human animation: HumanDiT Gan et al. (2025) introduces a pose-guided DiT for high-fidelity, long-duration synthesis; DreamActor-M1 Luo et al. (2025) improves multi-scale modeling and facial detail; OmniHuman-1 Lin et al. (2025) enhances scalability via mixed-data training. UniAnimate-DiT Wang et al. (2025b) fine-tunes Wan2.1 using LoRA Hu et al. (2022) for realistic animation, while RealisDance-DiT Zhou et al. (2025) proposes two fine-tuning strategies to accelerate convergence and preserve model priors. X-UniMotion Song et al. (2025) proposes a unified, identity-agnostic latent representation for whole-body motion. Inspired by these, we also build upon a large-scale video DiT to generate high-fidelity, long-duration human motion.

## 2.3 LONG VIDEO GENERATION IN HUMAN IMAGE ANIMATION

Generating long videos remains a challenge in video generation. Despite extensive efforts, existing methods remain limited in human image animation due to task-specific challenges. HumanDiT simply uses the final frame of one segment as the initial frame for the next. DreamActor-M1 and DreamActor-H1 Wang et al. (2025a) enable long video generation by using the final latent of the current segment as the initial latent for the subsequent segment. UniAnimate-DiT adopts the overlapped sliding window strategy to support long video generation. Although these methods can generate long videos, they remain prone to the forgetting and drifting issues described in FramePack Zhang & Agrawala (2025). Figure 2 shows long video generation results using UniAnimate-DiT with the three strategies. The sliding window introduces transition artifacts; final-frame initialization distorts identity and causes abrupt changes (e.g., frame 324 to 325); and latent-state propagation accumulates errors, leading to severe quality degradation.

## 3 METHOD

Our method generates high-fidelity, long-form human motion videos from a single reference image and a driving video. The framework is illustrated in Figure 3. We provide a detailed description of our approach as follows: In Section 3.1, we overview the technical preliminaries. Next, Section 3.2 introduces our hybrid guidance signals. In Section 3.3, we present the Position Shift Adaptive Module for long-duration synthesis. Finally, Section 3.4 details the pose alignment model and data augmentation strategy.

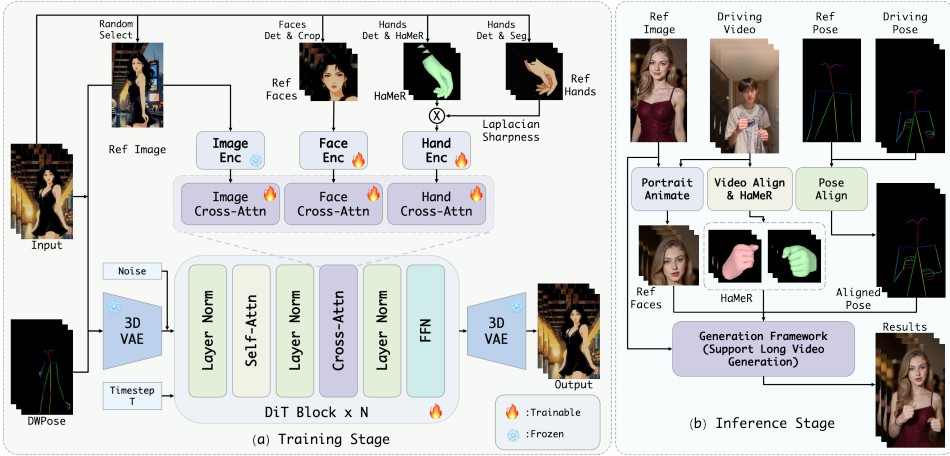

Figure 3: Overview of the proposed method. Given a single reference image and a driving video, our framework generates high-fidelity, long-duration human motion videos with strong temporal coherence and detail preservation.

## 3.1 PRELIMINARIES

**Flow Matching.** Flow Matching Lipman et al. (2022) is a simulation-free framework for training Continuous Normalizing Flows (CNFs), offering an alternative formulation of diffusion models via ODEs with stable training dynamics and equivalence to maximum likelihood. During training, given a reference human image and hybrid guidance signals $g$ extracted from the ground-truth video, our model is optimized to learn the reverse transformations. The objective is formulated using a mean squared error (MSE) loss between the model output and $v_t$:

$$\mathcal{L} = \mathbb{E}_{z_0, z_1, g, t} \left\| u\left(z_t, g, t; \theta\right) - v_t \right\|^2, \tag{1}$$

where $z_0 \sim \mathcal{N}(\mathbf{0}, \mathbf{I})$ is the initial noise, $z_1$ denotes the latent of the training sample, and $z_t$ is linearly interpolated between $z_0$ and $z_1$. The target velocity is $v_t = \frac{dz_t}{dt} = z_1 - z_0$, and $u\left(z_t, g, t; \theta\right)$ denotes the model's prediction parameterized by $\theta$, conditioned on $g$.

**Diffusion Transformers.** In this work, we build upon Wan2.1 as our base model. As illustrated in Figure 3, we utilize the pre-trained 3D Variational Autoencoder (VAE) Kingma et al. (2013) from Wan2.1 to encode both the input human image and the 2D pose skeleton sequences. Additionally, we incorporate a face encoder to extract face embeddings as implicit appearance guidance, along with a hand encoder to capture detailed hand features, enabling the generation of realistic hand structures.

## 3.2 HYBRID GUIDANCE SIGNALS

Most methods rely solely on a single reference image for video generation, often struggling with temporal consistency and visual fidelity in multi-scale and multi-view settings. Inspired by DreamActor-M1, we introduce hybrid implicit guidance signals using multi-frame facial and HaMeR sequences to enhance expressiveness and hand realism. Additionally, we introduce a sharpness guidance factor that enables the model to generate clear hand textures even when the driving video exhibits motion blur.

**Implicit Facial Representation.** To enhance facial generation quality, we introduce an implicit facial representation module. We initialize a face encoder $E_f$ with the appearance feature extractor $\mathcal{F}$ from LivePortrait Guo et al. (2024), differing from DreamActor-M1 which extracts motion features. During training, faces are detected, cropped, and resized to $256 \times 256$, forming a tensor $F \in \mathbb{R}^{t \times 3 \times 256 \times 256}$. The encoder $E_f$, followed by an MLP, maps each frame to a facial latent code $f \in \mathbb{R}^{t \times c}$, which is injected into the DiT blocks via cross-attention. At inference, we first animate the reference image using LivePortrait to generate a coherent facial sequence:

$$F = \mathcal{G}\left(\mathcal{W}\left(\mathcal{F}(I_{\text{ref}}), \mathcal{M}(I_{\text{ref}}), \mathcal{M}(V_{\text{dri}})\right)\right), \tag{2}$$

where $\mathcal{F}$, $\mathcal{M}$, $\mathcal{W}$, and $\mathcal{G}$ denote the appearance extractor, motion extractor, warper, and decoder of LivePortrait, respectively. $I_{\text{ref}}$ and $V_{\text{dri}}$ are the reference image and driving video. This animated

sequence serves as a dynamic facial reference, providing consistent and expressive appearance guidance throughout the generation process.

**Implicit Hand Representation.** While few methods focus on hand generation, approaches like RealisDance use 3D hand poses (e.g., from HaMeR) to improve hand quality. Similarly, we introduce an implicit hand representation for richer, appearance-aware guidance. We initialize a hand encoder $E_h$ with HaMeR's Vision Transformer (ViT) backbone. HaMeR is a state-of-the-art 3D hand gesture estimation method that leverages the scaling-up capabilities of Transformers which is trained on large-scale hand datasets, enabling it to capture more detailed hand structures than DWPose Yang et al. (2023). During training, hands are detected, cropped, and segmented using SAM2 Ravi et al. (2024) to isolate the hand regions. Then, we use the frozen ViT to extract HaMeR sequences. The encoder $E_h$, followed by an MLP, maps each HaMeR image to a latent code $h \in \mathbb{R}^{t \times c}$, which is injected into the DiT blocks via cross-attention, enabling fine-grained modeling of hand appearance and pose for improved realism and coherence.

**Laplacian Sharpness Factor.** Hand motion blur is common in training videos, but pose estimators (e.g., DWPose, HaMeR) fail to capture its intensity. Training with sharp guidance and blurry targets causes the model to overfit to blurred hand appearances. To address this, during training we compute the Laplacian Sharpness of hands and modulate the HaMeR sequence with it, implicitly conveying blur level to the model. At inference, we set the sharpness guidance factor to $1.0$ to encourage clear hand generation.

### 3.3 Long Video Generation

The Wan2.1 model can only generate videos with 81 frames. As a result, generating longer videos without degrading visual fidelity or breaking temporal coherence presents a substantial challenge. To address this, we employ the time-aware position shift fusion module proposed by Sonic and further modify the input structure of the DiT backbone for video frames. Sonic's fusion is efficient with no overhead. However, due to the special architecture of Wan2.1, which is designed to support joint training on both video and image data, direct application of this strategy may lead to abrupt transitions. Specifically, Wan2.1 processes input $V \in \mathbb{R}^{(1+T) \times H \times W \times 3}$ by splitting it into $1 + T/4$ chunks, where the first frame is spatially compressed alone to better handle the image data. The Wan-VAE then maps these to latent shape $[1 + T/4, H/8, W/8]$. However, for shifted windows $[s, e]$ with $s \neq 0$, the temporal structure becomes inconsistent: the first latent token derives from four frames, while others may come from single frames—violating the model's input design. This mismatch with the original design can easily lead to abrupt transitions. To resolve this, we split the video into $T/4$ uniform chunks and encode each into a single latent token, ensuring structural consistency across all segments. We term this adaptation the Position Shift Adaptive Module (Figure 4). Our full long-video generation pipeline is outlined in Algorithm 1. This approach enables high-quality, temporally coherent generation of videos with arbitrary duration.

### 3.4 Robust Pose Alignment

To handle inter-personal body shape variations, DreamActor-M1 and UniAnimate-DiT align skeletal proportions by scaling the driver's bone lengths to match the reference. However, this method can be unstable in edge cases, often requiring iterative refinement and manual tuning. We propose a data augmentation strategy and a dedicated alignment model to robustly mitigate shape discrepancies across identities.

**Data Augmentation Strategy.** To encourage the model to focus on relative skeletal motion rather than absolute positional cues, we apply independent random cropping and scaling to the reference image, pose sequences, and ground-truth video. Specifically, at each data loading step, we sample two independent parameter sets $(c_1, s_1)$ and $(c_2, s_2)$ for cropping and scaling. The reference image $I$ and ground-truth video $V$ are transformed using $(c_1, s_1)$, while the pose sequence $P$ is augmented with $(c_2, s_2)$. This mismatched augmentation decouples appearance geometry from motion dynamics, promoting robustness to spatial variations.

**Alignment and Smoothness Model.** To improve skeleton alignment between driving and reference subjects, we train an alignment and smoothness model adapted from SmoothNet Zeng et al. (2022). Using SMPL-X Pavlakos et al. (2019), we first get 3D body parameters and project pose sequences

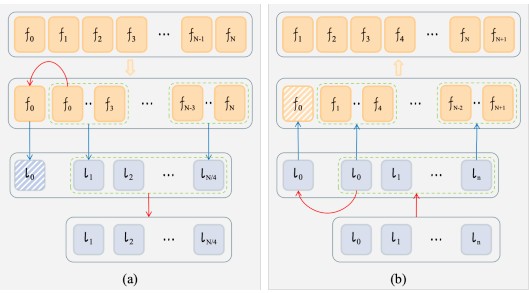

Figure 4: Workflow of the Position Shift Adaptive Module. (a) shows the encoding process; (b) shows the decoding process.

---

**Algorithm 1** Long Video Generation

---

**Input**: Human image $I$, driving video $V$ with length $L$, pretrained model $M(\cdot)$ for sequence length $f$, denoising steps $T$, position-shift offset $\alpha < f < \frac{L}{4}$, pretrained VAE encoder $E(\cdot)$ and decoder $D(\cdot)$.
**Preprocess**: Get reference latent code $r$ from $I$, get guidance signals $g$ from $V$ and $I$, get driving pose sequences $P$ from $V$.

1: Let $l = \frac{L}{4}$, initial noisy latent $z^{[0,l]}$.
2: $P' \leftarrow [P[0]; \, P]$. // Copy the first frame and concatenate.
3: $p' = E(P')$. // Obtain the latent code.
4: $p \leftarrow p'[1 :]$ // Discard the latent code of the first frame.
5: Initialize accumulated shift offset $\alpha_\Sigma = 0$.
6: **for** $t = T$ to 1 **do**
7:     // Denoising loop
8:     // Start from new position for each timestep.
9:     Initialize start point $s = \alpha_\Sigma$, end $e = s + f$, processed length $n = 0$.
10:     **while** $n < l$ **do**
11:         // Sequence loop
12:         $z_{t-1}^{[s,e]} = M(z_t^{[s,e]}, g^{[s,e]}, p^{[s,e]}, r, t)$
13:         $s \leftarrow s + f, e \leftarrow e + f, n \leftarrow n + f$. // Move to next clip non-overlapping.
14:         **if** $s > l$ **or** $e > l$ **then**
15:             $s \leftarrow s\%l, e \leftarrow e\%l$. // Padding circularly.
16:         **end if**
17:     **end while**
18:     $\alpha_\Sigma \leftarrow \alpha_\Sigma + \alpha$. // Accumulate shift offset.
19: **end for**
20: $z_0^{[0,l+1]} \leftarrow [z_0^{[0,l]}[0]; \, z_0^{[0,l]}]$. // Copy and concatenate.
21: $F^{[0,L+1]} = D(z_0^{[0,l+1]})$. // Obtain the generated video.
22: $F^{[0,L]} \leftarrow F^{[0,L+1]}[1 :]$ // Discard the first frame.
23: **return** Generated video $F^{[0,L]}$.

---

$P$ matching the subject's body shape. We then modify the shape parameters $\beta$ to apply spatial scaling (horizontal and vertical stretching), generating new sequences $P'$ with identical motion but altered body proportions. During training, a randomly selected frame from $P$ serves as the reference $P_{\text{ref}}$, and the model learns to map $(P_{\text{ref}}, P')$ to $P$.

## 4 EXPERIMENT

### 4.1 EXPERIMENTAL SETUPS

We train on a custom dataset comprising diverse scenarios such as sports and public speaking, ensuring broad coverage of human motion and facial expressions. In total, the training set comprises approximately 53,000 video segments with a cumulative duration of ∼170 hours. We initialize all weights from the pretrained Wan2.1-I2V-14B-720P model and train the model on 32 NVIDIA A100 GPUs for 20,000 steps at a resolution of $720 \times 1280$. We use the AdamW optimizer with a learning rate of $1 \times 10^{-5}$. During inference, we apply classifier-free guidance (CFG) Ho & Salimans

Table 1: Quantitative comparisons of video quality with state-of-the-art methods. The best results are in **bold**, and the second-best are in underlined.

| Methods | VBench↑ | | | | | FID↓ | FVD↓ | SSIM↑ | PSNR↑ | LPIPS↓ |
|---|---|---|---|---|---|---|---|---|---|---|
| | Subject Consist | BG Consist | Temporal Flicker | Motion Smooth | Aesthetic Quality | | | | | |
| *TikTok dataset* | | | | | | | | | | |
| RealisDance | 93.16 | 94.38 | 96.85 | 98.37 | 49.00 | 59.95 | 437.00 | 0.88 | 34.71 | 0.19 |
| DisPose | 94.81 | 95.08 | 97.44 | 97.96 | 48.76 | 51.21 | 293.37 | 0.89 | 32.49 | 0.17 |
| UniAnimate-DiT | 95.47 | 94.65 | 97.70 | 98.87 | 47.69 | **42.66** | 215.90 | **0.93** | **68.18** | **0.12** |
| RealisDance-DiT | 94.46 | 94.59 | 96.91 | 98.32 | 49.08 | 68.83 | 427.78 | 0.85 | 31.08 | 0.23 |
| Ours | **95.56** | **95.53** | **97.89** | **98.90** | **49.37** | 45.85 | **172.16** | 0.92 | 39.22 | 0.13 |
| *Our dataset* | | | | | | | | | | |
| RealisDance | 91.57 | 92.93 | 96.36 | 97.77 | 51.48 | 84.93 | 335.95 | 0.86 | 35.61 | 0.20 |
| DisPose | 92.31 | 93.68 | 97.55 | 98.55 | 53.43 | 79.32 | 289.17 | 0.83 | 32.98 | 0.20 |
| UniAnimate-DiT | 91.41 | 93.53 | 97.46 | 98.45 | 52.23 | 62.26 | 223.81 | 0.89 | 38.88 | 0.14 |
| RealisDance-DiT | 92.42 | 94.03 | 97.54 | 98.47 | 51.96 | 73.72 | 287.60 | 0.88 | 34.78 | 0.18 |
| Ours | **92.54** | **94.49** | **98.56** | **98.71** | **53.76** | 59.39 | **195.57** | **0.91** | **42.10** | **0.11** |

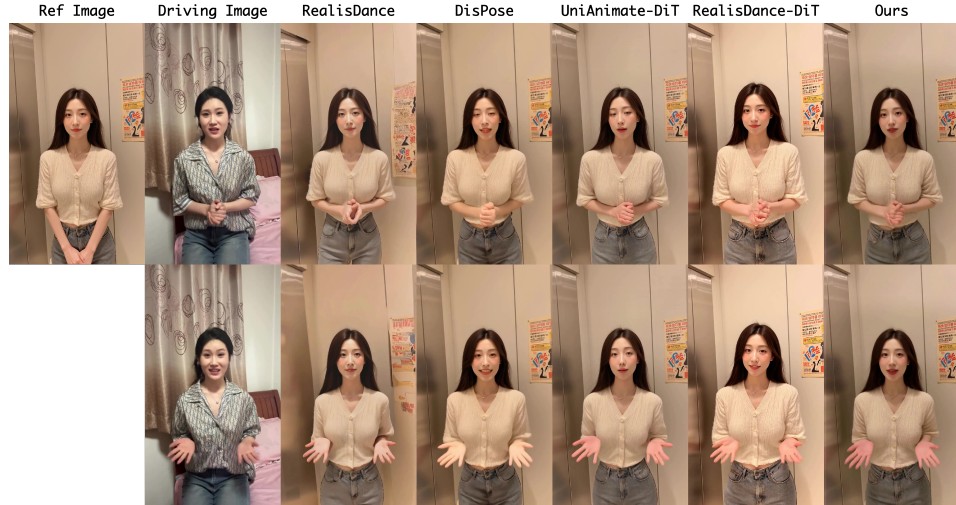

Figure 5: Qualitative comparison with other competing methods.

(2022) with a guidance scale of 5.0. We evaluate the superiority of our method using several widely adopted metrics from prior work, including FID, SSIM, LPIPS, PSNR, and FVD. Additionally, we adopt VBench Huang et al. (2024), including metrics such as subject and background consistency, temporal flickering, motion smoothness, dynamic degree, and aesthetic quality, enabling a holistic assessment of fidelity, coherence, and perceptual quality.

## 4.2 RESULTS AND ANALYSIS

We perform the qualitative and quantitative evaluation of our method with SOTA human image animation methods, including RealisDance, DisPose, UniAnimate-DiT, and RealisDance-DiT. The concurrent work OmniHuman-1, DreamActor-M1, and DreamActor-H1 have not yet released their code and models, so we cannot make comparisons. We use the TikTok dataset and our collected dataset as the test set.

**Quantitative Results.** As shown in Table 1, our method achieves the best results across all evaluation metrics on our test dataset, with some metrics outperforming all compared methods by a significant margin. On the TikTok test set, our method obtains the highest scores on the V-Bench metrics. It is worth noting that the test data may have been included in UniAnimate-DiT's training set, whereas our training data excludes all samples from the TikTok dataset. Despite this potential advantage for UniAnimate-DiT and the resulting performance gap in some metrics, our method still achieves the best results in terms of both VBench and FVD among all compared approaches, with all other metrics ranking second-best, demonstrating strong overall competitiveness.

Table 2: Hand Pose Evaluation and User Study. The best results are in **bold**, and the second-best are in underlined.

| Methods | Hand Pose Evaluation | | | | User Study | | |
|---|---|---|---|---|---|---|---|
| | PA-MPJPE ↓ | AUC$_J$ ↑ | AUC$_V$ ↑ | F@5 ↑ | VC ↑ | MA ↑ | AF ↑ |
| RealisDance | 15.45 | 0.58 | 0.59 | 0.26 | 1.80 | 2.70 | 2.40 |
| DisPose | 28.46 | 0.41 | 0.42 | 0.12 | 3.10 | 3.10 | 2.20 |
| UniAnimate-DiT | 21.48 | 0.45 | 0.47 | 0.15 | 3.30 | 3.00 | 3.10 |
| RealisDance-DiT | 16.47 | 0.57 | 0.58 | 0.20 | 3.30 | 3.40 | 3.00 |
| Ours | **14.69** | **0.70** | **0.71** | **0.52** | **4.00** | **4.00** | **4.10** |

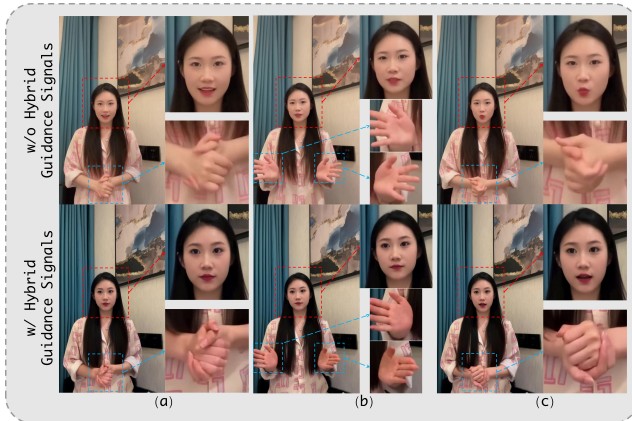

Figure 6: Ablation on Hybrid Guidance Signals.

**Qualitative Results.** As shown in Figure 5, RealisDance exhibits background distortions; DisPose fails to preserve facial details; UniAnimate-DiT shows deviations in body shape and pose, along with noticeable hand deformities; and RealisDance-DiT suffers from blurred hands and inconsistent shoulder width. In contrast, our method produces sharper hand textures when motion blur exists in the driving video, improved facial fidelity, and better body shape alignment with the reference. Since human image animation involves dynamic content, static frames cannot fully reflect motion naturalness and temporal coherence. Therefore, we provide full video comparisons in the supplementary material.

**Hand Pose Accuracy.** We conducted a hand pose evaluation to demonstrate our method's superiority in hand generation. Specifically, we use HaMeR to extract MANO parameters from both the driving and generated videos, then compute 3D hand keypoints and vertex coordinates. We evaluate 3D joint and mesh accuracy using PA-MPJPE, AUC$_J$, AUC$_V$, and F@5mm. As shown in Table 2, our method significantly outperforms the others across all metrics.

**User Study.** To further validate the effectiveness of our proposed method, we conducted a user study with 50 participants, who rated the videos using a 5-point Mean Opinion Score (MOS) scale across three critical dimensions: Video Consistency, Motion Accuracy, and Appearance Fidelity. As shown in Table 2, our method achieves the highest scores on all metrics, demonstrating its superiority in subjective evaluation.

### 4.3 ABLATION STUDIES

**Ablation on Hybrid Guidance Signals.** We trained models with and without hybrid guidance signals, respectively, and evaluated their effectiveness in Section 3.2. As shown in Figure 6, we compare the results generated by the two models from the same driving video. The model without guidance signals produces videos with poorer facial and hand clarity, physically implausible textures during hand crossing, and reduced hand integrity. In contrast, the model with guidance signals generates more realistic and complete faces and hands, with physically plausible gestures and higher facial consistency.

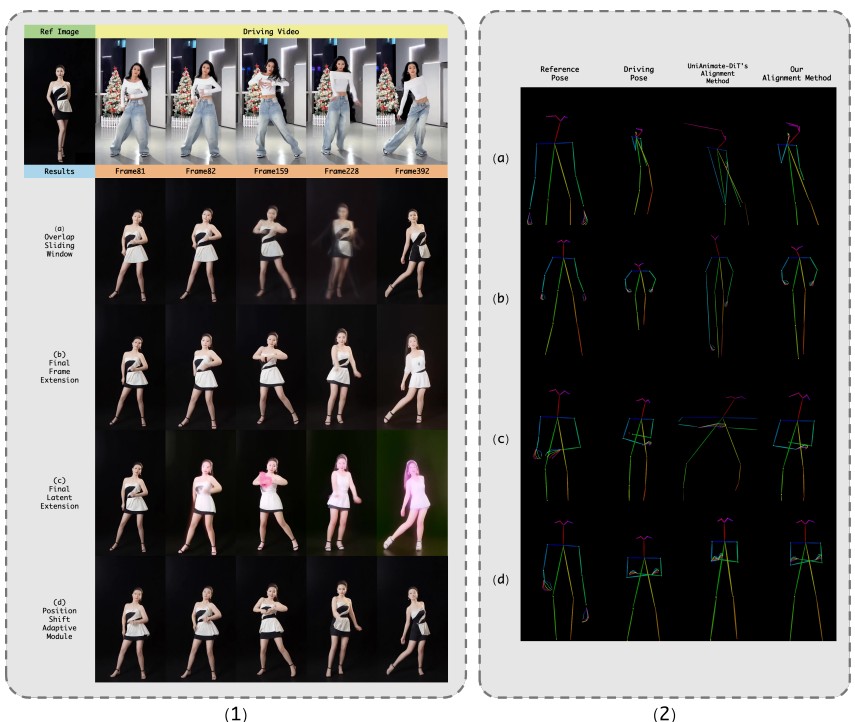

Figure 7: Ablation on Long Video Generation.

**Ablation on Long Video Generation.** We ablate the Position Shift Adaptive module by comparing a model without the modifications in Section 3.3 against our full method. We evaluate three long-video generation strategies from Section 2.3: (1) overlapped sliding window, (2) final frame extension, and (3) final latent extension. As shown in Figure 7(1), the sliding window introduces artifacts at transition regions (e.g., frames 159, 228) due to frame overlap. Final frame extension fails to preserve identity and causes abrupt changes at segment boundaries (e.g., frames 81–82). Final latent extension suffers from error accumulation, degrading output quality over time. In contrast, our method generates temporally coherent and identity-preserving videos, maintaining high visual fidelity and smoothness throughout.

**Ablation on Pose Alignment.** We analyze the effectiveness of our alignment model in multi-scale and multi-view scenarios. As shown in Figure 7(2)(a), UniAnimate-dit's alignment method results in misalignment of head and hand regions under large pose angle differences. Under significant distance variations, it leads to unnatural neck and hand bone lengths, as well as hand crossing artifacts (Figure 7(2)(b)(d)). Even in typical cases, it introduces overall skeleton scaling artifacts (Figure 7(2)(c)). In contrast, our alignment model achieves accurate and robust alignment across all three challenging scenarios.

## 5 CONCLUSION

We present a novel DiT-based framework for high-fidelity and long-duration human image animation. To overcome the limitations of existing methods in generating fine-grained details and extended video sequences, we introduce three key components: hybrid implicit guidance signals to enhance facial and hand motion realism and fidelity, a Position Shift Adaptive Module that enables flexible generation of videos with arbitrary duration, and a robust preprocessing pipeline incorporating data augmentation and skeleton alignment to mitigate identity-related shape variations. Extensive experiments demonstrate that our approach achieves state-of-the-art performance in both visual quality and temporal coherence. To the best of our knowledge, our method is the first to demonstrate human image animation videos exceeding one minute in duration. By effectively balancing detail preservation, motion naturalness, and scalability, our method advances the capabilities of human animation systems and shows strong potential for real-world applications requiring high-quality, long-form content generation.

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
