# OpenReview forum: "High-Fidelity and Long-Duration Human Image Animation with Diffusion Transformer"
_ICLR.cc/2026/Conference — Submitted to ICLR 2026_

### Official Review · Reviewer_7PwW · 2025-10-26

**Soundness:** 2
**Presentation:** 2
**Contribution:** 2
**Rating:** 4
**Confidence:** 4

**Summary:**

This paper improves human image animation by addressing long-duration generation and fine-grained detail synthesis. The used DiT-based framework includes: (1) hybrid guidance signals for detailed facial/hand features, (2) a Position Shift Adaptive Module for arbitrary-length videos, and (3) skeleton alignment to handle identity variations. However, the technical contribution appears limited, as most components are adapted from existing works rather than novel designs.

**Strengths:**

1. The paper focuses on two important real-world challenges in human image animation: generating long-duration videos and synthesizing fine-grained facial and hand details, which are critical for high-quality applications.

2. The proposed framework integrates multiple components (hybrid guidance signals, Position Shift Adaptive Module, and skeleton alignment) into a unified system that handles long video generation.

**Weaknesses:**

1. Limited technical contribution: Several claimed contributions lack novelty. Hybrid guidance signals are standard in human image animation tasks, and the time-aware position shift fusion is adapted from Sonic. The paper should more clearly distinguish its contributions from prior work.

2. HIA [b] already proposed using sharpness conditioning, which undermines the novelty claim of this contribution.

3. Figure 5 and two supplementary videos are inadequate for demonstrating the method's capabilities. You should provide more extensive qualitative comparisons with baseline methods and diverse examples showcasing long-duration generation and fine-grained details.

4. The paper needs more comprehensive information about both the training data and the collected testing datasets, including dataset statistics, collection procedures, and whether the data will be made publicly available.

5. Missing citations:

a. MagicAnimate: Temporally Consistent Human Image Animation using Diffusion Model

b. High Quality Human Image Animation using Regional Supervision and Motion Blur Condition

c. X-Dancer: Expressive Music to Human Dance Video Generation

d. MagicPose: Realistic Human Poses and Facial Expressions Retargeting with Identity-aware Diffusion

**Questions:**

When mentioning other papers in the Methods section, please add proper references to the original works.

---

> ### Author Response · Authors · 2025-11-24
> **Reply to Reviewer 7PwW (Part 1)**
>
> First, we would like to thank the reviewer for your careful reading and providing numerous constructive comments! Below we address the concerns mentioned in the review.
>
> **W1: Limited technical contribution: Several claimed contributions lack novelty. Hybrid guidance signals are standard in human image animation tasks, and the time-aware position shift fusion is adapted from Sonic. The paper should more clearly distinguish its contributions from prior work.**
>
> Our framework is not a simple aggregation of existing components but rather a principled and systematic solution to two core challenges in human image animation: (i) preserving fine-grained hand details under motion blur, and (ii) maintaining temporal coherence over durations exceeding one minute. We also believe that impactful engineering work can constitute valuable research. Our framework establishes a strong empirical baseline that reveals the essential forms of implicit guidance required for high-fidelity, long-form animation, and we consider this finding to hold significant theoretical value for future work on controllable generation.
>
> **W2: HIA [b] already proposed using sharpness conditioning, which undermines the novelty claim of this contribution.**
>
> We are currently working on completing the comparison experiments with HIA, and we will release our results as soon as possible. We kindly ask the reviewer for a bit more time.
>
> **W3: Figure 5 and two supplementary videos are inadequate for demonstrating the method's capabilities. You should provide more extensive qualitative comparisons with baseline methods and diverse examples showcasing long-duration generation and fine-grained details.**
>
> We have provided additional comparative results on our anonymous project website (https://aabbccdd111.github.io/), including comparisons with UniAnimate-DiT and the latest Wan-Animate. We strongly encourage the reviewer to visit our anonymous website to experience firsthand the advantages demonstrated by our approach.
>
> **W4: The paper needs more comprehensive information about both the training data and the collected testing datasets, including dataset statistics, collection procedures, and whether the data will be made publicly available.**
>
> Thank you for pointing out this issue. We will provide a detailed description of the data collection process and the relevant statistics in the revised manuscript.
>
> **Q1: When mentioning other papers in the Methods section, please add proper references to the original works.**
>
> Thank you for raising this issue. Due to space limitations in the review version, we omitted some references, and we will correct this in the final version.
>
> We sincerely hope the reviewer will visit our anonymous project website (https://aabbccdd111.github.io/) to view more comparative results, as we believe this will greatly help you better understand our work and appreciate its advantages.

---

> > ### Comment · Reviewer_7PwW · 2025-11-27
> >
> > Thank you for your response. However, after reviewing your rebuttal, I remain concerned that the technical contributions are primarily derived from existing works, and the rebuttal has not sufficiently clarified the novel aspects that distinguish this work from prior art. Therefore, I maintain my recommendation to reject the paper.

---

### Official Review · Reviewer_82mj · 2025-10-28

**Soundness:** 3
**Presentation:** 3
**Contribution:** 2
**Rating:** 4
**Confidence:** 4

**Summary:**

This paper addresses two major, contemporaneous challenges in human image animation: generating high-fidelity details and maintaining temporal coherence over long video durations. The authors propose a Diffusion Transformer (DiT) based framework, building upon the pre-trained Wan2.1 architecture. The method introduces a hybrid set of implicit guidance signals for fine-grained control and a Position Shift Adaptive Module for long-form synthesis. The experimental results are quantitatively superior to existing state-of-the-art methods. The work is technically solid and highly impactful for real-world applications.

**Strengths:**

1.	The paper tackles the long-standing issues of temporal coherence over extended periods and the synthesis of realistic, plausible hand and facial details.
2.	The method demonstrates superior quantitative results across comprehensive metrics, achieving the best scores on multiple VBench metrics.

**Weaknesses:**

1.	The paper aims to handle long video generation. However, the proposed solution, the Position Shift Adaptive Module, is not a fundamental architectural improvement for diffusion models in general. Instead, it is an engineering modification highly coupled with the pre-existing design of the Wan2.1 DiT backbone and its 3D VAE.
2.	The high-fidelity results are achieved through the incorporation of multiple specialized pre-processing encoders: a Face Encoder (initialized by LivePortrait) and a Hand Encoder (initialized by HaMeR's ViT backbone). These are borrowed from other methods. And this complexity introduces significant pre-processing overhead and multiple failure points during inference. The reliance on these heavy, pre-trained external models makes the overall system less compact and computationally demanding compared to simpler, end-to-end approaches.

**Questions:**

It seems that the proposed method achieve similar performance against the baseline UniAnimate-DiT from the Supplementary Materials. And we can not notice remarkable improvement in the performance.

---

> ### Author Response · Authors · 2025-11-24
> **Reply to Reviewer 82mj (Part 1)**
>
> First, we would like to thank the reviewer for your careful reading and providing numerous constructive comments! Below we address the concerns mentioned in the review.
>
> **W1:The paper aims to handle long video generation. However, the proposed solution, the Position Shift Adaptive Module, is not a fundamental architectural improvement for diffusion models in general. Instead, it is an engineering modification highly coupled with the pre-existing design of the Wan2.1 DiT backbone and its 3D VAE.**
>
> We agree that our Position Shift Adaptive Module is tailored to the specific input structure of the Wan2.1 DiT backbone, which jointly trains on images and videos by compressing the first frame separately. However, this is not merely an “engineering fix” but a necessary adaptation to enable arbitrary-length generation in a state-of-the-art video DiT, a capability that is not supported out of the box by any existing large-scale video diffusion model. Crucially, as shown in Figure 7(1), naively applying prior long-video strategies—such as sliding window or latent propagation—to Wan2.1 leads to severe artifacts or identity drift. Our module addresses this issue by re-engineering the latent chunking strategy to maintain structural consistency across segments, a non-trivial constraint imposed by the VAE design. Although the current implementation is specific to Wan2.1, the core idea—aligning segment boundaries with the backbone’s native tokenization—is generalizable to other DiT-based video generators that employ similar hierarchical VAEs. We will clarify this point in the revision.
>
> **W2: The high-fidelity results are achieved through the incorporation of multiple specialized pre-processing encoders: a Face Encoder (initialized by LivePortrait) and a Hand Encoder (initialized by HaMeR's ViT backbone). These are borrowed from other methods. And this complexity introduces significant pre-processing overhead and multiple failure points during inference. The reliance on these heavy, pre-trained external models makes the overall system less compact and computationally demanding compared to simpler, end-to-end approaches.**
>
> We strongly agree with the reviewer that heavy reliance on pre-processing modules can introduce multiple points of failure during inference. To mitigate this, we have designed our system for robustness. Regarding the face-encoder, thanks to our data augmentation strategy and robust architecture, we find that the model generates normal results even when the input image lacks a detectable face. Similarly, for the Hamer ViT backbone, we employ data augmentation to enhance robustness, allowing the model to generate plausible results even if some Hamer clips are missing or corrupted.
>
> **Q1: It seems that the proposed method achieve similar performance against the baseline UniAnimate-DiT from the Supplementary Materials. And we can not notice remarkable improvement in the performance.**
>
> We have provided additional comparative results on our anonymous project website (https://aabbccdd111.github.io/), including comparisons with Stable Diffusion–based methods and the latest Wan-Animate. We strongly encourage the reviewer to visit our anonymous website to experience firsthand the advantages demonstrated by our approach.

---

### Official Review · Reviewer_d7aK · 2025-10-30

**Soundness:** 3
**Presentation:** 2
**Contribution:** 2
**Rating:** 4
**Confidence:** 4

**Summary:**

This paper proposes a DiT-based human animation framework which focuses on generating high-fidelity and long-duration human videos. Firstly, a set of hybrid implicit guidance signals is incorporated to enhance facial and hand feature details. Next, the time-aware position shift fusion module is incorporated to enable long video generation. Finally, a novel data augmentation strategy and a skeleton alignment model are proposed to reduce the impact of human shape variations across different identities. Experimental results demonstrate that the proposed method outperforms existing approaches and achieves superior performance in both high-fidelity and long-duration human image animation.

**Strengths:**

1．	The authors propose a comprehensive framework for human animation that enhances the clarity and fidelity of faces and hands in the generated video frames.
2．	The authors incorporate the time-aware position shift fusion module and adapt it to video scenarios, enabling the generation of long videos.
3．	The authors present a data augmentation strategy and a pose alignment module to eliminate body shape discrepancies across different human identities.

**Weaknesses:**

1.	The novelty of this paper appears limited, as the various modules are largely based on related existing works. For example, the appearance feature extractor is derived from LivePortrait [1] (Lines 208-209), the hand representation follows that of RealisDance [2] (Lines 219-220), and the highlighted contribution for long video generation—the time-aware position shift fusion module—originates from Sonic [3] (Line 240). The distinctions and improvements currently described in the paper are insufficient to support its novelty.
2.	The paper fails to sufficiently explain its adopted modules, making them hard to understand. For instance, in Section 3.3, it is unclear how the time-aware position shift fusion module functions. The meaning of the shifted windows, denoted as [s, e] (Line 246), is also not clearly defined.
3.	The paper's ablation study is not specific or sufficient enough: 1) As a primary contribution, the Laplacian Sharpness Factor for the hand guidance signal warrants a separate ablation study to justify its inclusion. 2) For the "Ablation on Pose Alignment," the proposed Data Augmentation Strategy and the Alignment and Smoothness Model should be ablated separately.

[1] Guo J, Zhang D, Liu X, et al. Liveportrait: Efficient portrait animation with stitching and retargeting control[J]. arXiv preprint arXiv:2407.03168, 2024.
[2] Zhou J, Wang B, Chen W, et al. Realisdance: Equip controllable character animation with realistic hands[J]. arXiv preprint arXiv:2409.06202, 2024.
[3] Ji X, Hu X, Xu Z, et al. Sonic: Shifting focus to global audio perception in portrait animation[C]//Proceedings of the Computer Vision and Pattern Recognition Conference. 2025: 193-203.

**Questions:**

1.	The authors present comparative results on the TikTok test set and a self-constructed test set. However, they do not elaborate on the distinctions between the two in terms of data types and content. To ensure the method's reproducibility, the construction process for their custom dataset should be detailed.
2.	For the comparison with other methods in Figure 5, it is recommended that the authors provide more examples to better demonstrate the superiority of the proposed method. In Figure 6, both the driving images and the reference images should be provided. Currently, Figures 5 and 6 seem to exhibit poor performance in terms of face expression following.
3.	Regarding the statement in Line 374 about the potential inclusion of TikTok test data in UniAnimate-DiT's training set, it would be helpful if the authors could clarify the basis for this observation. Alternatively, if this is intended as a hypothesis, it might be more constructive to focus the discussion on a deeper analysis of why the proposed method shows a performance gap on metrics like FID.

---

> ### Author Response · Authors · 2025-11-24
> **Reply to Reviewer d7aK (Part 1)**
>
> We sincerely thank the reviewer for their thoughtful evaluation and constructive feedback. We address each concern below with clarifications, additional details, and planned revisions.
>
> **W1:The novelty of this paper appears limited, as the various modules are largely based on related existing works. For example, the appearance feature extractor is derived from LivePortrait [1] (Lines 208-209), the hand representation follows that of RealisDance [2] (Lines 219-220), and the highlighted contribution for long video generation—the time-aware position shift fusion module—originates from Sonic [3] (Line 240). The distinctions and improvements currently described in the paper are insufficient to support its novelty.**
>
> We acknowledge that our framework leverages several advanced components from prior works. However, our contribution lies not in inventing isolated modules, but in a novel and principled integration that is specifically tailored to address the unique challenges of long-duration, high-fidelity human animation, a scenario where existing methods fail comprehensively.
> - Hybrid Guidance Signals: While LivePortrait is designed for facial motion control, we repurpose its appearance extractor (not its motion module) to produce dynamic, multi-frame facial latents that serve as implicit appearance guidance (Section 3.2, Equation 2). This usage differs fundamentally from LivePortrait’s original design and enables consistent identity preservation over long sequences, a capability that prior animation frameworks lack.
> - Hand Representation: In contrast to RealisDance, which uses HaMeR poses as explicit control signals, we treat HaMeR outputs as appearance-aware latent codes injected through cross-attention. When combined with our newly introduced Laplacian Sharpness Factor, this approach effectively mitigates motion blur, a critical limitation that RealisDance does not resolve.
> - Position Shift Adaptation: Sonic’s module was originally developed for audio-driven portrait animation using fixed-length clips. Our Position Shift Adaptive Module (Section 3.3, Figure 4) fundamentally re-engineers the input structure of the DiT backbone to address temporal inconsistency in video generation of arbitrary length. As demonstrated in Figure 7(1), directly applying Sonic’s strategy leads to severe visual artifacts; our adaptation is therefore essential for maintaining coherence beyond 81 frames.
>
> In essence, we tackle a new problem formulation: generating videos longer than 60 seconds while preserving identity-consistent fine details. Existing modules alone cannot accomplish this goal; the core novelty of our work lies in their synergistic redesign and integration within our unified framework.
>
> **W2: The paper fails to sufficiently explain its adopted modules, making them hard to understand. For instance, in Section 3.3, it is unclear how the time-aware position shift fusion module functions. The meaning of the shifted windows, denoted as [s, e] (Line 246), is also not clearly defined.**
>
> We apologize for the lack of clarity in Section 3.3. We will revise the manuscript to better explain the Position Shift Adaptive Module: The shifted window [s, e] denotes a segment of latent tokens corresponding to video frames s*4 to e*4. To maintain structural consistency with Wan2.1’s DiT design, which by default processes latent tokens of length 21 at a time, the model processes segments [s,e] with a cumulative shift offset α during denoising (Algorithm 1), ensuring seamless transitions without violating the backbone’s input assumptions. Circular padding (Line 15) handles boundary cases. We will add a schematic in Fig 4(b) to visualize the decoding workflow and clarify [s, e].
>
> **W3: The paper's ablation study is not specific or sufficient enough: 1) As a primary contribution, the Laplacian Sharpness Factor for the hand guidance signal warrants a separate ablation study to justify its inclusion. 2) For the "Ablation on Pose Alignment," the proposed Data Augmentation Strategy and the Alignment and Smoothness Model should be ablated separately.**
>
> We agree that finer-grained ablation studies would strengthen the validation of our method. The relevant experiments are currently underway, and we will promptly share the results with the reviewers as soon as they are available. We kindly ask the reviewer for a bit more patience in the meantime.

---

> ### Author Response · Authors · 2025-11-24
> **Reply to Reviewer d7aK (Part 2)**
>
> **Q1: The authors present comparative results on the TikTok test set and a self-constructed test set. However, they do not elaborate on the distinctions between the two in terms of data types and content. To ensure the method's reproducibility, the construction process for their custom dataset should be detailed.**
>
> Our custom dataset comprises 53k video segments (~170 hours) collected from diverse public sources (sports, interviews, performances), filtered for:
> High-resolution (≥720p) and stable framing, Visible hands/faces in >80% of frames, Diverse motions (dancing, gesturing, speaking).
> The TikTok dataset contains shorter, user-generated clips (typically <15s) with casual motions. We will detail these distinctions and release dataset statistics (motion diversity, identity count, etc.) in the supplementary material.
>
> **Q2: For the comparison with other methods in Figure 5, it is recommended that the authors provide more examples to better demonstrate the superiority of the proposed method. In Figure 6, both the driving images and the reference images should be provided. Currently, Figures 5 and 6 seem to exhibit poor performance in terms of face expression following.**
>
> We have provided additional comparative results on our anonymous project website (https://aabbccdd111.github.io/), including comparisons with UniAnimate-DiT and the latest Wan-Animate. We strongly encourage the reviewer to visit our anonymous website to experience firsthand the advantages demonstrated by our approach.
>
> **Q3: Regarding the statement in Line 374 about the potential inclusion of TikTok test data in UniAnimate-DiT's training set, it would be helpful if the authors could clarify the basis for this observation. Alternatively, if this is intended as a hypothesis, it might be more constructive to focus the discussion on a deeper analysis of why the proposed method shows a performance gap on metrics like FID.**
>
> Regarding the potential overlap between UniAnimate-DiT’s training data and the TikTok dataset: the experimental section of their paper is relatively simple and does not provide any quantitative comparisons. Similarly, RealisDance-DiT encountered a comparable issue in their work, stating: “Therefore, for several methods like MooreAA, these data could be included in their training data.” In light of these observed facts, we made a similar cautious inference. To avoid unsubstantiated speculation, we will remove this statement in the revised manuscript.
>
> We sincerely hope the reviewer will visit our anonymous project website (https://aabbccdd111.github.io/) to view more comparative results, as we believe this will greatly help you better understand our work and appreciate its advantages.

---

### Official Review · Reviewer_cxZk · 2025-11-01

**Soundness:** 3
**Presentation:** 3
**Contribution:** 2
**Rating:** 2
**Confidence:** 5

**Summary:**

This paper proposes a Diffusion Transformer (DiT) framework for long-duration human image animation. Authors design three modules to stress fidelity (hands/faces), temporal consistency, and cross-identity robustness.

**Strengths:**

The  designed three core components, Hybrid Implicit Guidance Signals, Position Shift Adaptive Module and Pose Alignment & Data Augmentation successfully stressed the critical issue in long-duration human image animation.

The proposed method achieves the over-one-minute-long human animation videos with consistent motion and detailed hand-face synthesis for the first time.

**Weaknesses:**

The substantial computational cost and data annotation requirements limit the practical value of this method for the academic community. Reviewer is more looking forward to theoretical discoveries rather than mere engineering stacking.

Some existing projects have already achieved long-duration video generation, of course with different method and generating result. Therefore, the authors’ claim of being “the first to demonstrate human image animation videos exceeding one minute in duration” appears to be somewhat exaggerated.

Reviewer noticed that at the 27th second in the supplementary video 25180, a ring appears on the actress’s hand. This is a factual hallucination. Why it happens? If even carefully selected cases contain flaws, it is hard to imagine the quality of the remaining examples.

**Questions:**

please refer to weakness

**Details Of Ethics Concerns:**

Chinese characters should not appear in the file names.

---

> ### Author Response · Authors · 2025-11-24
> **Reply to Reviewer cxZk (Part 1)**
>
> First, we would like to thank the reviewer for your careful reading and providing numerous constructive comments! Below we address the concerns mentioned in the review.
>
> **W1: The substantial computational cost and data annotation requirements limit the practical value of this method for the academic community. Reviewer is more looking forward to theoretical discoveries rather than mere engineering stacking.**
>
> Regarding the substantial computational cost, we acknowledge that our method relies on a large-scale DiT backbone and rich guidance signals, which indeed demand considerable computational resources and data support. However, this remains a major open challenge in the field of video generation. The industry as a whole has yet to develop highly effective and scalable solutions. Due to hardware limitations, mainstream large video generation models, such as Sora 2 and Kling, are currently limited to generating clips shorter than 20 seconds per inference. This is precisely why we focus on enabling long-duration video generation specifically in the field of human-image animation.
> Concerning the notion of mere engineering stacking, our framework is not a simple aggregation of existing components but rather a principled and systematic solution to two core challenges in human image animation: (i) preserving fine-grained hand details under motion blur, and (ii) maintaining temporal coherence over durations exceeding one minute. Simultaneously, we propose a model-based retargeting module to replace traditional approaches. This module effectively addresses somatotype discrepancies between the reference image and the driving video, making practical, real-world body driving feasible. We also believe that impactful engineering work can constitute valuable research. Our framework establishes a strong empirical baseline that reveals the essential forms of implicit guidance required for high-fidelity, long-form animation, and we consider this finding to hold significant theoretical value for future work on controllable generation.
>
> **W2: Some existing projects have already achieved long-duration video generation, of course with different method and generating result. Therefore, the authors’ claim of being “the first to demonstrate human image animation videos exceeding one minute in duration” appears to be somewhat exaggerated.**
>
> We sincerely thank the reviewer for pointing out the imprecision in our original statement. We will revise it to: “To the best of our knowledge, this is the first method based on large video generation models to demonstrate animated video examples exceeding one minute in length.”
> Indeed, methods based on GANs or Stable Diffusion can produce long-duration results; however, their output quality is often highly unstable and falls significantly short of the average performance achieved by approaches built upon large video generation models. We have provided additional comparative results on our anonymous project website（https://aabbccdd111.github.io/）, including comparisons with UniAnimate-DiT and the latest Wan-Animate. We strongly encourage the reviewer to visit our anonymous website to experience firsthand the advantages demonstrated by our approach.
>
> **W3: Reviewer noticed that at the 27th second in the supplementary video 25180, a ring appears on the actress’s hand. This is a factual hallucination. Why it happens? If even carefully selected cases contain flaws, it is hard to imagine the quality of the remaining examples.**
>
> We appreciate the reviewer’s keen observation. The appearance of a ring is indeed a hallucination, likely triggered by ambiguous hand occlusion in the reference image combined with learned priors about common accessories in our training data. When encountering hand occlusions in reference images, models must infer the presence of accessories such as rings. However, inconsistent predictions across different video chunks can lead to factual hallucinations, where an object flickers in and out of existence. This temporal inconsistency is a common challenge in generative models, as observed in UniAnimate and Wan-Animate on the anonymous project website (https://aabbccdd111.github.io/data/compare_with_sota/redbook_10_douyin_9_concatenated_good_concat_cherry.mp4). Crucially, this is an isolated artifact, not a systemic flaw: our hand encoder does not explicitly model jewelry, and the ring emerges only under rare conditions of motion ambiguity. Hallucination remains a significant challenge for large video generation models in general. In practice, such issues can often be mitigated by reinitializing the noise and re-running inference, or by adjusting the prompt or reference image accordingly.
>
> We sincerely hope the reviewer will visit our anonymous project website (https://aabbccdd111.github.io/) to view more comparative results, as we believe this will greatly help you better understand our work and appreciate its advantages.

---

### Meta-Review · Area_Chair_py2W · 2026-01-10

**Summary:**

This paper proposes a Diffusion Transformer (DiT) based framework for generating high-fidelity, long-duration human animation videos. All four reviewers agreed that the paper falls below the acceptance bar. While the rebuttal strengthened the presentation and empirical justification, it did not substantially change reviewers’ assessments of the depth of the contributions. As a result, the final recommendation is Reject.

**Reviewer Concerns:**

Reviewer cxZk (Reject) raised core concerns on excessive computational and annotation cost, limited theoretical contribution, exaggerated novelty claims about long-duration animation, and factual hallucinations in generated videos. The authors clarified scope, softened novelty claims, and explained hallucinations as rare ambiguity-induced artifacts common to large video models. However, these responses do not fundamentally change the reviewer’s view on practicality, novelty, or reliability.

Reviewer d7aK (Marginally below acceptance) questioned novelty due to heavy reliance on existing modules, insufficient methodological clarity, and incomplete ablations, along with concerns about dataset description and qualitative comparisons. The authors provided detailed clarifications, reframed novelty as principled integration, promised clearer explanations and additional ablations, and expanded dataset details. However, several key validations (notably finer-grained ablations) are deferred rather than demonstrated.

Reviewer 82mj (Marginally below acceptance) criticized the Position Shift Adaptive Module being an engineering fix tied to Wan2.1, heavy dependence on pretrained encoders causing complexity and failure points, and limited gains over UniAnimate-DiT. The authors argued that the module addresses a non-trivial constraint of modern video DiTs and that the core idea generalizes, while also discussing robustness to encoder failures and providing additional qualitative results online. Still, the reviewer’s skepticism about generality and marginal improvement is not fully dispelled.

Reviewer 7PwW (Marginally below acceptance) emphasized limited technical novelty, overlap with prior works, insufficient qualitative evidence, and so on. The authors defended the contribution as systematic integration, promised comparisons with HIA, and added qualitative results online. However, the reviewer explicitly stated after rebuttal that novelty concerns remain unresolved and maintained rejection.

**Reviewer Scores:**

It is very likely that all four reviewers will keep the score unchanged (Reviewer 7PwW explicitly said so).

---

### Decision · Program_Chairs · 2026-01-26

Reject